# Influence of a Two-Dimensional Growth Mode on Electrical Properties of the GaN Buffer in an AlGaN/GaN High Electron Mobility Transistor

**DOI:** 10.3390/ma15176043

**Published:** 2022-09-01

**Authors:** Volkan Esendag, Peng Feng, Chenqi Zhu, Rongzi Ni, Jie Bai, Tao Wang

**Affiliations:** Department of Electrical and Electronic Engineering, The University of Sheffield, Mappin Street, Sheffield S1 3JD, UK

**Keywords:** unintentional doping, dislocations, GaN, AlGaN, electrical characterisation, capacitance-voltage, electrical breakdown

## Abstract

An extensive study has been conducted on a series of AlGaN/GaN high electron mobility transistor (HEMT) samples using metalorganic vapour phase epitaxy, to investigate the influence of growth modes for GaN buffer layers on device performance. The unintentional doping concentration and screw dislocation density are significantly lower in the samples grown with our special two-dimensional (2D) growth approach, compared to a widely-used two-step method combining the 2D and 3D growth. The GaN buffer layers grown by the 2D growth approach have achieved an unintentional doping density of 2 × 10^14^ cm^−3^, two orders lower than 10^16^ cm^−3^ of the GaN samples grown using a conventional two-step method. High-frequency capacitance measurements show that the samples with lower unintentional doping densities have lower buffer leakage and higher breakdown limits. This series of samples have attained sub-nA/mm leakages, a high breakdown limit of 2.5 MV/cm, and a saturation current density of about 1.1 A/mm. It indicates that our special 2D growth approach can effectively lessen the unintentional doping in GaN buffer layers, leading to low buffer leakage and high breakdown limits of GaN/AlGaN HEMTs.

## 1. Introduction

GaN-based band gap semiconductors are alluring for power electronics due to a number of major advantages in comparison with other III-V semiconductors [1,2,3,4,5,6,7,8,9,10,11,12,13], such as the inherently high breakdown electric fields which are disputed to be anywhere between 3 and 3.7 MV/cm, the saturation carrier velocity of 2.5 × 10^7^ ms^−1^ and the intrinsic electron mobility around 900–1250 cm^2^ V^−1^ s^−^^1^ at room temperature which can be further enhanced up to 2100 cm^2^ V^−1^ s^−1^ through the formation of an AlGaN/GaN heterostructure leading to high electron mobility transistors (HEMTs).

As a result, research into high electron mobility transistors (HEMT), from the very first inception in III-arsenide material system [14,15], has been a cornerstone of III-nitride semiconductor material research for high-frequency, high power, and high-temperature applications [16,17,18,19,20,21,22]. In order to maximise the exploitation of the major advantages and maintain a performance faithful to the figure of merit of the material, it is necessary to minimise the buffer leakage of GaN-based HEMTs, one of the fundamental issues that prevent a HEMT structure from naturally excellent electronic performance. It is generally regarded that dislocations lead to the formation of unintentional donor/acceptor traps, in particular, the resultant deep level traps [23,24], providing an effective path for current leakage [23,25,26]. However, the relationship between the fundamental issue and a growth mode is unclear, which is of paramount importance for further improving the electrical properties of the GaN buffer in AlGaN/GaN HEMTs.

It is well-known that a classic two-step growth approach has been widely used for GaN grown on sapphire by using metalorganic vapour phase epitaxy (MOVPE) techniques. This approach consists of the initial deposition of a thin GaN or AlN nucleation layer at a low temperature (LT) and the preparation of a thick GaN buffer layer at a high temperature (HT) prior to the growth of any further device structures. For the two-step growth method, small islands on a nanometre scale initially form as a result of a subsequent annealing process underwent on the LT nucleation layer, followed by a gradual coalescence process. Finally, a flat surface can be obtained. It means that the growth of GaN on sapphire initially follows a three-dimensional (3D) growth mode and then takes a two-dimensional (2D) growth mode. However, it is a great challenge to obtain a semi-insulating GaN buffer layer which is required for GaN electronics by using this two-step approach [27,28,29]. Very recently, we employed a high-temperature (HT) AlN buffer technique for the growth of an AlGaN/GaN HEMT structure on sapphire [30,31,32,33,34], demonstrating an extremely low off-state buffer leakage current of down to 1 nA/mm at 1000 V [35]. Unlike the classic two-step growth method, our high-temperature AlN buffer technique leads to a 2D growth mode throughout the whole growth process. Furthermore, our GaN on sapphire obtained by this 2D growth mode exhibits an extremely narrow full width at half maximum (FWHM) of the on-axis X-ray diffraction (XRD) rocking curve along (0002) direction [31], which implies an extremely low screw dislocation density.

In this study, a systematic investigation has been carried out on a series of HEMT structures grown by using our HT AlN buffer techniques, where the AlN buffer was prepared under different growth conditions. Detailed on-axis and off-axis XRD measurements have been performed on these samples to examine their screw dislocation densities, which have been found to depend on the growth conditions, in particular, the V/III ratio during the AlN buffer growth. A capacitance-voltage (CV) technique, a powerful non-destructive tool which allows for working acceptably with low doping concentrations, has been employed to assess the unintentional doping densities in the GaN buffer layers in these samples [36]. A combination of both XRD and CV measurements has confirmed that the 2D growth approach leads to a significant reduction in screw dislocation density, which massively reduces the unintentional doping levels in the GaN buffers of our AlGaN/GaN HEMT structures by using the HT AlN buffer technique.

## 2. Experimental Section 

In this work, four HEMT samples, which were all grown on c-plane (0001) sapphire substrates using MOVPE, have been studied, labelled Sample A to D. Figure 1a schematically illustrates a block diagram of the epitaxial structure for the four HEMT samples. It is worth highlighting that Sample A was grown by a modified two-step method, with a 25 nm GaN nucleation layer grown at 550 °C, followed by a HT GaN buffer layer grown at a temperature a little bit lower than usual. The decreased growth temperature of the GaN buffer aims to form a semi-insulating GaN buffer layer by increasing carbon doping for compensation of the unintentional n-type doping. The other three samples, Sample B, C, and D, were all grown by our HT AlN buffer approach, which started with an AlN buffer layer grown at high temperature and followed by a HT GaN buffer layer. This HT-AlN growth approach initiates with a 2D growth, which has been demonstrated in previous work [35], as atomic force microscope (AFM) images revealed parallel and straight terraces but without any screw dark spots on the sample surface which indicates 2D-only growth. The subsequent structure after the 1.5 μm GaN buffer layer is identical for all samples, composed of a 1 nm AlN spacer layer and a 30 nm Al_0.2_Ga_0.8_N layer, as demonstrated in Figure 1a.

Table 1 shows layer-by-layer growth conditions of the individual samples as a function of temperature, pressure, and the flow rates of precursors. For both the AlN buffer layer and the GaN buffer layer, compared to Sample A, the V/III flow rate ratios of Sample B, C, and D were grown with much higher V/III flow rate ratios to encourage the 2D growth. 

Moreover, the GaN buffer was grown at a higher pressure of 225 Torr for Sample A and Sample B, but at a lower pressure of 175 Torr for Sample C and Sample D. In addition, only for Sample D, the GaN buffer layer was grown at a bit higher temperature. The growth conditions for the aforementioned HEMT structures, a thin GaN layer, an AlN spacer and an Al_0.2_Ga_0.8_N layer subsequently grown, are all identical for the four samples. 

After the growth, Schottky Barrier Diode (SBD)-like devices are fabricated on the HEMT epi-wafers for investigation of unintentional doping in the GaN buffer layers, as shown in Figure 1b. First, UV photolithography and inductively coupled plasma (ICP) etching were used to expose the main GaN buffer layer all around the sample except the contact sites. Ti/Al/Ni/Au metals with thicknesses of 20/70/20/55 nm were deposited as the Ohmic contact using thermal evaporator. The contacts were then rapidly thermally annealed at 800 °C for 30 s in N_2_ ambient. Next, Si_3_N_4_ is deposited using plasma-enhanced chemical vapour deposition (PECVD) for passivation, with the intent of suppressing surface state conduction and parasitic leakage paths. This is important for the purpose of gauging the inherent leakage from the buffer layer. Finally, the Si_3_N_4_ on top of the contacts is removed by ICP etching for measurements.

Moreover, HEMT devices are also fabricated on the HEMT epi-wafers as shown in Figure 1c. The fabrication process is described as below: A 300 nm depth mesa is etched down to define the active region for the HEMT by ICP etching, then the metal stack of Ti/Al/Ni/Au (20/150/30/80 nm) is deposited and then annealed at 800 °C for 30 s in N_2_ ambient for 30 s in order to form Ohmic contacts for the source and drain of the HEMT. Finally, a Ni/Au (50/150 nm) alloy is deposited in order to form a Schottky gate for the HEMT. 

## 3. Results and Discussion

Previously, we have attempted to prove the reduction or elimination of auto-C doping by adjusting the buffer growth parameters, through low-temperature photoluminescence (PL) measurements [35]. Due to that, the deep states induced by auto-C doping increase the emission wavelength, centred at around 550 nm, as opposed to the expected 357 nm, the PL emission spectra exhibit the expected 357 nm corroborating to eliminated C-doping (not shown here).

In order to obtain the dislocation information, X-ray diffraction (XRD) measurements are performed on the four samples, using a Bruker X-ray diffractometer with a 1.54 Å Cu-Kα tube. Figure 2a,b record ω rocking curves of the four samples along the (002) lattice axis and along the (102) axis, which can reflect densities of the screw and mixed dislocations in the epitaxial samples, respectively. Table 2 shows the corresponding full widths at half of the maximum (FWHM) and calculated dislocation densities. Compared to Sample A, the screw dislocation densities of Sample B, C, and D are one order lower, which means that the screw dislocation density is decreased with the 2D growth being enhanced. The mixed dislocation density, however, does not follow a corresponding decrease and is a lot more complicated, suggesting that the edge dislocation density does not follow this trend. Given that the buffer leakage paths are mainly affected by screw dislocations, it implies that the 2D growth is effective in suppressing breakdown limits.

The voltage-current (I-V) characteristics are measured on the four HEMT devices using a Keithley source measure unit, connected to a PC for remote testing and data extraction, and a probe station equipped with an optical microscope. The biases are from −20 V to 20 V. The measurements were conducted on a few devices for each sample to ensure consistency in results. Moreover, the transistor characteristics are measured through two sweeping voltages applied to the gate and drain, of −9 to 1 V and 0 to 10 V, respectively.

Figure 3a displays an I-V plot of the device with the highest saturation current, i.e., Sample B. Figure 3b demonstrates its corresponding consistent transconductance curve. The transistor Wg is 10 µm; Lg and Ldg are 2 µm and Lds is 6 µm. The drain-source voltage for the transconductance measurements is 10 V. A high current rating of about 1.1 A/mm is obtained for Sample B. Though there is a report on higher current densities, it employed intentionally doped barriers and different substrates [26]. It is noted that the record I-V curve is not quite smooth, which could be related to donor traps. Nevertheless, a high current density has been obtained in a HEMT device which is achieved through our special 2D growth and without an intentional doping.

The capacitance characteristics are captured on the SBD structures by a Keysight E4980A LCR (inductance-capacitance-resistance) meter, which is calibrated for an open circuit prior to measurements. These are performed at 1 MHz to ensure the frequency is large enough to both reduce series reactances and improve the reliability of the results for small-signal frequency of operation. Capacitance data included here is of the parallel equivalent circuit model. Voltage biases of −20 to 0 V have been chosen to roughly correspond to depletion voltage regimes. Figure 4a shows a collective of representative capacitances at reverse bias measured from −20 to 0 V for all the samples. Figure 4b displays unintentional doping concentrations which are obtained by integrating the resultant capacitances over the voltage range, using the equation as follows [37]:(1)$$n\left(2D\right)=\int_{V_{T}}^{V}\frac{CdV}{\left(qA\right)}$$
where *C* is the measured capacitance per point of the devices; *V_T_* is the lower voltage threshold used to integrate the capacitance data; *V* is the upper limit (in our case, *V* = 0); *q* is the electron charge (1.6 × 10^−19^ *C*) and *A* is the active area of the device. Because we have no Schottky contacts, the inter-contact area is used instead. The end result as a 2D charge density can be extrapolated onto a 3D space to give the final result for the calculated unintentional doping concentration per unit volume rather than sheet area.

Sample A has the highest capacitance, as high as the pF sort of range, which corresponds to the highest unintentional doping density. In addition, its increase in capacitance over the given voltage range is also the most significant among these samples, which is an indicator of a depletion regime for the unintentional doping. Compared to Sample A, the other three samples successively show a slower rise in capacitance with the voltage as well as lower starting capacitance. Though, when zoomed in, the capacitance increase for Sample A within this regime is overshadowed by the instrument’s innate noise, the integration method is resilient enough to be able to still calculate the doping density with reasonable accuracy.

Table 3 lists the measured performances of the four samples in terms of five quantities: Overall calculated unintentional doping concentrations obtained from the contacts’ capacitance characteristics; screw dislocation density; buffer leakage current at 20 V; breakdown field limits [35], and forward saturation current density at V_GS_ = 1 V. It is found that the unintentional doping concentrations of Sample B, C and D are one order or even two orders lower than that of Sample A. It indicates that the 2D growth efficiently reduces the unintentional doping, which can be further lessened by a lower pressure during the buffer growth. It results in a strong link between the screw dislocation density and the unintentional doping concentration, i.e., the screw dislocation density in the sample is always higher when the doping concentration is higher. It further confirms that screw dislocations lead to the formation of unintentional donor/acceptor traps, and can be significantly prevented by the 2D growth.

Furthermore, as shown in Table 3, the samples with lower screw dislocation densities are found to have lower leakages and higher breakdown field limits, such as Sample B, Sample C, and Sample D, due to lessened leakage paths and an increase of channel off-resistance. Sample D, with the lowest screw dislocation density, has achieved a well-recorded breakdown characteristic of 2.5 MV/cm, exceeding the breakdown limits of even intentionally C-doped buffer structures, which are limited to 2 MV/cm due to deep acceptor traps and the impurities-introduced extra dislocations [35,38]. The measurements were performed on similar buffer structures separated by 3 µm in a special high-voltage permitted laboratory with source measure units with sweeps up to 1000 V. Fluorinert has been used to prevent aerial breakdown above 300 V. The end result for the breakdown field was obtained by converting the applied voltage to the field strength and the current to mA/mm. However, it is worth mentioning that the samples with lower unintentional doping densities attain lower saturation currents, which is not good for device performance. Though Sample D is the best sample in terms of screw dislocation density, leakage, and breakdown limit, it is also the worst in terms of full device saturation current density. A modulation and delta doping has been reported to be used in c-plane GaN/AlGaN HEMTs to achieve extremely high current densities [26]. It is expected that the modulation doping can be employed for the growth of Sample D to induce both great on-state current densities and low off-state current densities with high breakdown voltage limits. Nevertheless, in the absence of intentional doping, Sample C offers the best compromise of a good saturation current density and a good performance with a high breakdown limit and low leakage. 

## 4. Conclusions

We have performed a comprehensive study on GaN/AlGaN HEMTs grown with a special 2D-growth approach, and made a comparison with a two-step method with a combination of 2D- and 3D-growth. It is found that the unintentional doping concentrations and screw dislocation densities are significantly lower in the 2D-grown samples compared to the two-step growth method, achieving a very low unintentional doping density of 2 × 10^14^ cm^−3^ and a screw dislocation density of 2.3 × 10^7^ cm^−2^. High-frequency capacitance measurements show that the samples grown with the 2D growth approach have much lower buffer leakage and higher breakdown limits, attaining a sub-nA/mm leakage, a high breakdown limit of 2.5 MV/cm, and a saturation current density of about 1.1 A/mm. It indicates that our special 2D-growth approach can effectively lessen the unintentional doping in the GaN buffer layers, leading to GaN/AlGaN HEMTs with low buffer leakage and high breakdown limits. 

## Figures and Tables

**Figure 1 materials-15-06043-f001:**
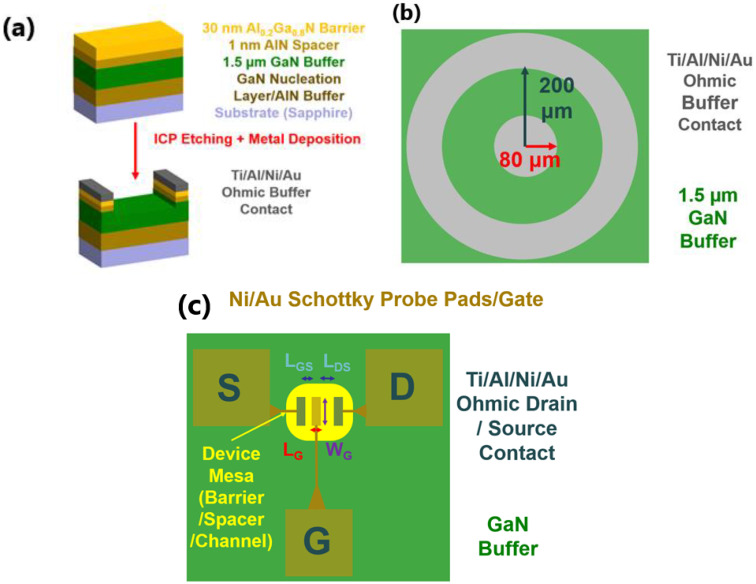
(**a**) A block diagram of the device structure. (**b**) A circular Schottky Barrier Diode pattern deposited only as Ohmic metal on the buffer. (**c**) A schematic configuration of the HEMT device.

**Figure 2 materials-15-06043-f002:**
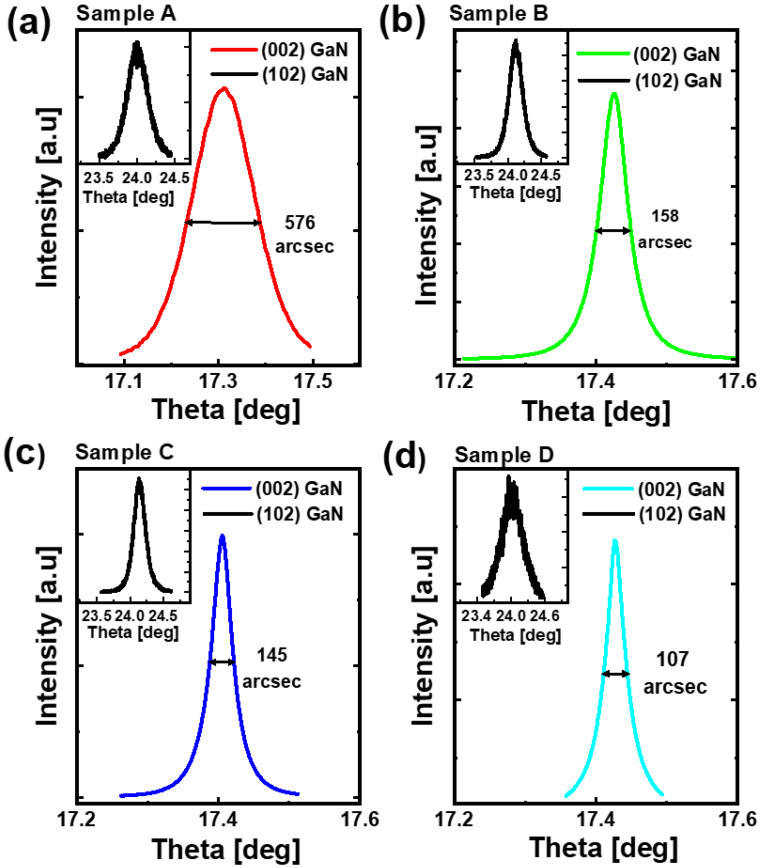
XRD measurements along the (002) axis and (102) axis for Sample A (**a**), Sample B (**b**), Sample C (**c**), and Sample D (**d**), respectively.

**Figure 3 materials-15-06043-f003:**
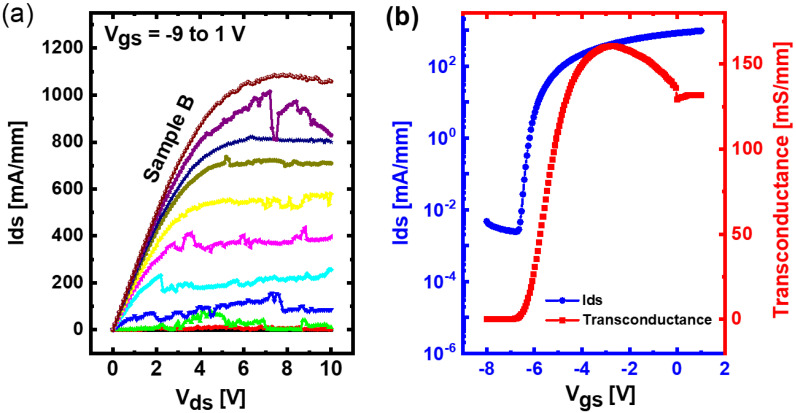
(**a**) Record I-V characteristics and (**b**) Consistent transconductance curves of a 10 µm gate length transistor for Sample B, the sample with the highest forward current performance.

**Figure 4 materials-15-06043-f004:**
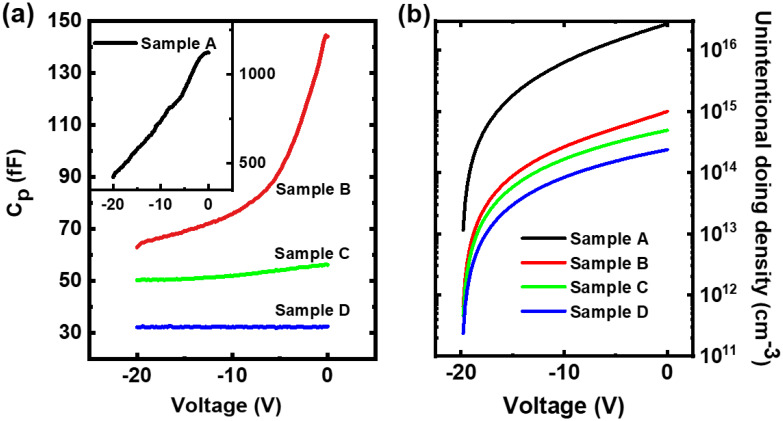
(**a**) Capacitance curves obtained for a Schottky Barrier Diode for all devices under negative bias. (**b**) The inferred unintentional doping curves for the four samples.

**Table 1 materials-15-06043-t001:** The growth parameters of the four samples.

Sample	Layer	Temperature [°C]	Pressure [Torr]	NH_3_ Flow Rate [sccm]	TMGa Flow Rate [sccm]	TMAl Flow Rate [sccm]
**A**	*AlGaN Barrier*	1100	75	5480	12.5	30
	*AlN Spacer*	1100	75	5480		30
	*GaN Channel*	1100	75	5480	12.5	
	1.5 μm *GaN Buffer*	1100	225	5480	65	
	25 nm *LT GaN Nucleation Layer*	550	65	1900	65	
**B**	*AlGaN Barrier*	1110	75	5480	12.5	30
	*AlN Spacer*	1110	75	5480		30
	*GaN Channel*	1100	75	5480	12.5	
	1.5 μm *GaN Buffer*	1110	225	5480	65	
	*HT-AlN Buffer*	1180	65	5480		180
**C**	*AlGaN Barrier*	1108	75	5480	12.5	30
	*AlN Spacer*	1108	75	5480		30
	*GaN Channel*	1108	75	5480	12.5	
	1.5 μm *GaN Buffer*	1108	175	5480	65	
	*HT-AlN Buffer*	1180	65	5480		180
**D**	*AlGaN Barrier*	1130	75	5480	12.5	30
	*AlN Spacer*	1130	75	5480		30
	*GaN Channel*	1130	75	5480	12.5	
	1.5 μm *GaN Buffer*	1130	175	5480	65	
	*HT-AlN Buffer*	1180	65	5480		180

**Table 2 materials-15-06043-t002:** The FWHMs of (002) and (102) XRD curves for each sample, and their correspondingly calculated dislocation densities.

Sample	(002) GaN FWHM [°]	Screw Dislocation Density [cm^−2^]	(102) GaN FWHM [°]	Mixed Dislocation Density [cm^−2^]
A	0.1599	6.7 × 10^8^	0.3005	8.6 × 10^9^
B	0.0438	5.0 × 10^7^	0.2363	5.3 × 10^9^
C	0.0402	4.2 × 10^7^	0.1948	3.6 × 10^9^
D	0.0296	2.3 × 10^7^	0.4429	1.9 × 10^10^

**Table 3 materials-15-06043-t003:** A summary of each sample’s performances in five categories: screw dislocation density, calculated unintentional doping concentration, buffer leakage for a 2 µm spacing, maximum saturation current density for a full HEMT device, and breakdown limit of the sample.

Sample	D_screw_ [cm^−2^]	Calculated N_unintentional_ [cm^−3^]	Buffer Leakage, 2 µm@20 V [pA]	HEMT Current Density, Lg = 10 µm [mA/mm]	Breakdown Limit of Buffer [MV/cm]
A	6.7 × 10^8^	1.0 × 10^16^	1136.4	1024	0.25
B	5.0 × 10^7^	1.0 × 10^15^	980.9	1086	0.6
C	4.2 × 10^7^	5.0 × 10^14^	615.0	920	1.9
D	2.3 × 10^7^	2.0 × 10^14^	41.7	582	2.5

## Data Availability

Not applicable.

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
