# Peer review of "Influence of a Two-Dimensional Growth Mode on Electrical Properties of the GaN Buffer in an AlGaN/GaN High Electron Mobility Transistor"

_materials, 2022, doi:10.3390/ma15176043_

Round 1

Reviewer 1 Report

The authors have conducted a very comprehensive study on the GaN buffer layer with various growth mode to investigate the influence on the device performance of AlGaN/GaN HEMTs. This is a very interesting topic and it will provide very important guidance on the buffer layer design for AlGaN/GaN HEMTs. The paper is well written and well organized. I think the paper should be accepted after clarifying the following two minor points.

1.       Keywords ‘Selective Epi Removal’ is not mentioned in the paper.

2.       It’s well known that carbon concentration in the buffer layer will affect the electrical properties of the HEMT devices, such as leakage current and breakdown voltage etc. I wonder if the authors could provide SIMS results for different types of samples to rule out the carbon concentration influence.

Author Response

We would like to thank you for their supportive and useful comments. Our response to the comments is detailed below by point-by-point. The changes have also been marked in track in the manuscript for your convenience.

Reviewer 2 Report

Paper is written in adequate quality. Study of the decrease  buffer leakage of the HEMT is still actual. I have few question or comments:

1) How do you know, that your HT-AlN approach start with 2D growth?
2) How thickness and growth condition was used to channel GaN layer on top buffer layer, which is important for HEMT properties?
3) What are the dimensions of the HEMT transistor: Lds, Ldg and Wg?
4) I-V curve of the HEMTs is not smooth (fig 3a). Do you obtained this characteristic after first measurement or after more cycle of the measurement? I aspect, that after repetitive measurement donor traps are filled and characteristic will be smother. How looks characteristic for other sample (A,C and D)?
5) How is value of the drain - source voltage (Vds) on the figure 3b? I think, that will be better show Ids in logarithmic scale, to see buffer leakage
6) Can you compare mopfhology  of samples surface with 2D a 3D growth mode by the AFM or SEM?
7) How was measured breakdown limit?

Author Response

We would like to thank you for your supportive and useful comments. Our response to the comments is detailed below by point-by-point. The changes have also been marked in track in the manuscript for your convenience.
